# Design of carbon supports for metal-catalyzed acetylene hydrochlorination

Selina K. Kaiser [1,4], Ivan Surin[1,4], Ana Amorós-Pérez[2], Simon Büchele[1], Frank Krumeich [1], Adam H. Clark [3], Maria C. Román-Martínez [2], Maria A. Lillo-Ródenas[2] & Javier Pérez-Ramírez [1✉]

For decades, carbons have been the support of choice in acetylene hydrochlorination, a key industrial process for polyvinyl chloride manufacture. However, no unequivocal design criteria could be established to date, due to the complex interplay between the carbon host and the metal nanostructure. Herein, we disentangle the roles of carbon in determining activity and stability of platinum-, ruthenium-, and gold-based hydrochlorination catalysts and derive descriptors for optimal host design, by systematically varying the porous properties and surface functionalization of carbon, while preserving the active metal sites. The acetylene adsorption capacity is identified as central activity descriptor, while the density of acidic oxygen sites determines the coking tendency and thus catalyst stability. With this understanding, a platinum single-atom catalyst is developed with stable catalytic performance under two-fold accelerated deactivation conditions compared to the state-of-the-art system, marking a step ahead towards sustainable PVC production.

[1] Institute for Chemical and Bioengineering, Department of Chemistry and Applied Biosciences, ETH Zürich, Zürich, Switzerland. [2] Department of Inorganic Chemistry and Materials Institute (IUMA), University of Alicante, Alicante, Spain. [3] Paul Scherrer Institut, Villigen, PSI, Switzerland. [4] These authors contributed equally: Selina K. Kaiser, Ivan Surin. ✉email: jpr@chem.ethz.ch

Carbon nanomaterials are indispensable for a broad range of applications, ranging from energy conversion and storage to $CO_2$ capture and heterogeneous catalysis[1–5]. In the latter context, carbons have been used since decades as support for metal-based catalysts in thermo- and electrocatalysis[4,6], and are nowadays dominantly employed to stabilize nanostructured metal sites and isolated atoms and tune their coordination environments and reactivity[7–11]. A prominent example is acetylene hydrochlorination, a key technology for the manufacture of polyvinyl chloride (13 Mton y$^{-1}$), which fully relies on carbon-supported catalysts, including the commercial $HgCl_2$-based system[12,13], as well as competing more sustainable alternatives based on Au[14–21], Pt[22,23], and Ru[24,25]. In recent years, metal nuclearity and coordination effects have been subject to extensive research in this reaction, revealing Au(I)Cl$_x$, Pt(II)Cl$_x$ single atoms and RuO$_x$Cl$_y$ nanoparticles as the active sites of the aforementioned catalysts. On the contrary, the widely recognized key role of the carbon support on the catalytic activity and stability of metal-based hydrochlorination catalysts remains ambiguous[26–28]. While the commonly used activated carbons (ACs) themselves are virtually inactive in acetylene hydrochlorination (vinyl chloride yield, $Y$(VCM) < 2%) under typical operating conditions (GHSV($C_2H_2$) = 650 h$^{-1}$ and $T_{bed}$ = 473 K)[29], carbon is a key ingredient to generate highly performing metal-based and metal-free catalysts (e.g., N-doped carbons)[29–32]. Other matrices such as silica, alumina, zeolites, etc. render much less active systems ($Y$(VCM) < 5%)[18,19]. The underlying reasons for this stark contrast have been ascribed to several characteristics of carbon, in particular the: (i) high surface area, (ii) metal mobility, (iii) acetylene adsorption capacity, and (iv) oxygen functionalities[18,19,26–28,33]. However, no unequivocal correlations could be drawn between these properties and the activity and stability of metal-based hydrochlorination catalysts due to the difficulties in disentangling metal nanostructure and support effects. In fact, all previous studies were exclusively dedicated to Au/AC, containing poorly stabilized Au nanostructures, which are susceptible to nuclearity changes during synthesis and under reaction conditions[14,34,35]. Notably, Au single atoms may be effectively stabilized through suitable N-anchoring sites in the carbon matrix. However, the latter also promote coking, leading to accelerated catalyst deactivation[14]. This aspect, in combination with the intrinsic activity of nitrogen-doped carbon (NC) in acetylene hydrochlorination, renders functionalized carbons less attractive compared to their non-functionalized analogs for fundamental investigations and practical application. To enable a systematic investigation into the role of the latter for metal-based catalysts in acetylene hydrochlorination, exploiting the high stability of Pt single atoms on various supports[7,36,37], including carbon[23], offers an attractive solution.

Herein, we systematically assess the impact of the porous (i.e., amount and type of pores) and chemical properties (i.e., oxygen functionalities) of the carbon support on the activity and stability of Pt-, Ru-, and Au-based catalysts in acetylene hydrochlorination, by preserving the respective active metal sites. Combining precise material synthesis, quantitative evaluation, and kinetic analysis, coupled with in-depth characterization, we assess which properties of carbon are key to yield highly active acetylene hydrochlorination catalysts and which ones are responsible for coking and/or pore blockage to derive general performance descriptors. With this understanding, we unravel the metal-carbon interplay for Au, Pt, and Ru and design a Pt single-atom catalyst (SAC) with unparalleled stability at enhanced productivity under accelerated deactivation conditions compared to the state-of-the-art systems.

## Results

**Platform of Pt single-atom catalysts with defined carbon hosts.**
To systematically assess the roles of the carbon support in acetylene hydrochlorination, a comprehensive platform of Pt/C single-atom catalysts was synthesized, employing a broad range of amorphous carbons with varying porous properties (i.e., surface area, pore volume, pore structure) and oxygen content, while preserving the Pt-Cl single atom site (Fig. 1a). Due to the high stability of Pt on the O-sites of carbon, Pt/C SACs (1 wt% Pt) can be obtained via a simple incipient wetness impregnation of the carbon hosts with an aqueous solution of $H_2PtCl_6$, followed by drying at 473 K in static air (reaction temperature in acetylene hydrochlorination). Details on the synthesis of the carbon supports, notation of the Pt/C catalysts, and all characterization techniques employed, are provided in the Supplementary Methods and Supplementary Tables 1, 2[23]. The formation of Pt single atoms was visualized by aberration-corrected scanning transmission electron microscopy (STEM) and corroborated by the absence of characteristic reflections of Pt in the X-ray diffraction (XRD) patterns (Fig. 1b, Supplementary Figs. 1–6a). On the basis of extended X-ray absorption fine structure (EXAFS) analysis and X-ray photoelectron spectroscopy (XPS) the comparable chemical nature and coordination environment of the active metal sites in the Pt/C catalysts was confirmed (Fig. 1c, d, Supplementary Figs. 7–9, Supplementary Tables 3–6). Specifically, Pt-Cl and Pt-O/C coordination numbers of ~3 ± 0.3 and ~1 ± 0.3 were determined, respectively. Combining XRD and Raman analyses, the amorphous nature of carbon was identified for all Pt/C catalysts (Supplementary Fig. 2). Based on $N_2$ and $CO_2$ sorption (Fig. 1e, f, Supplementary Table 7), minimal alterations of the porous properties through the impregnation method were found, thus preserving the wide porosity spectrum within the Pt/C catalysts with respect to the previously reported SAC[23], denoted here as benchmark Pt/AC. The total oxygen content and functionalities present in the carbon supports and the Pt/C catalysts were analyzed via thermogravimetric analysis in He coupled to mass spectrometry (TGA-MS) to determine the amount of oxygen evolving as CO and $CO_2$ (Supplementary Table 8). Based on the evolution profile (i.e., area and position of evolution peaks) of the individual gases, one can quantify the oxygen functionalities and assess their nature. Generally, $CO_2$ evolves at lower temperatures and is related to the decomposition of acidic functionalities, such as carboxylic groups or lactones, as further validated by temperature-programmed desorption (TPD) of ammonia coupled to MS analysis (Supplementary Fig. 10). CO evolution occurs at higher temperatures and indicates the decomposition of basic or neutral groups, such as phenols, ethers, and carbonyls[38,39]. Upon impregnation of the carbon support with the aqueous $H_2PtCl_6$ solution, Pt(IV) is partially reduced to Pt(II) via redox processes, thereby altering the surface functionalization of carbon[40]. During the subsequent drying step in air, the carbon surface likely undergoes additional oxidation reactions, leading to further oxygen incorporation. Consequently, all Pt/C samples show a characteristic increase in the total oxygen content upon impregnation with the metal precursor solution (Supplementary Table 7, 10–20%) which is particularly enhanced for the initially most reduced carbons (lowest oxygen contents, Fig. 1d). To gradually decrease the oxygen content, while preserving a comparable carbon structure and porosity, the AC4 support was thermally treated in inert atmosphere at varying temperatures, from 573 K to 1173 K as indicated in the respective sample codes (Supplementary Table 1). As a further means to characterize the oxygen sites, the O1$s$ XPS spectra were fitted for C–O (e.g., ether, alcohol groups) and C=O contributions (e.g., ketone, lactone, and/or carbonyl groups, Supplementary Fig. 11). In line with the TPD-MS results, the total surface oxygen content varied among the catalysts

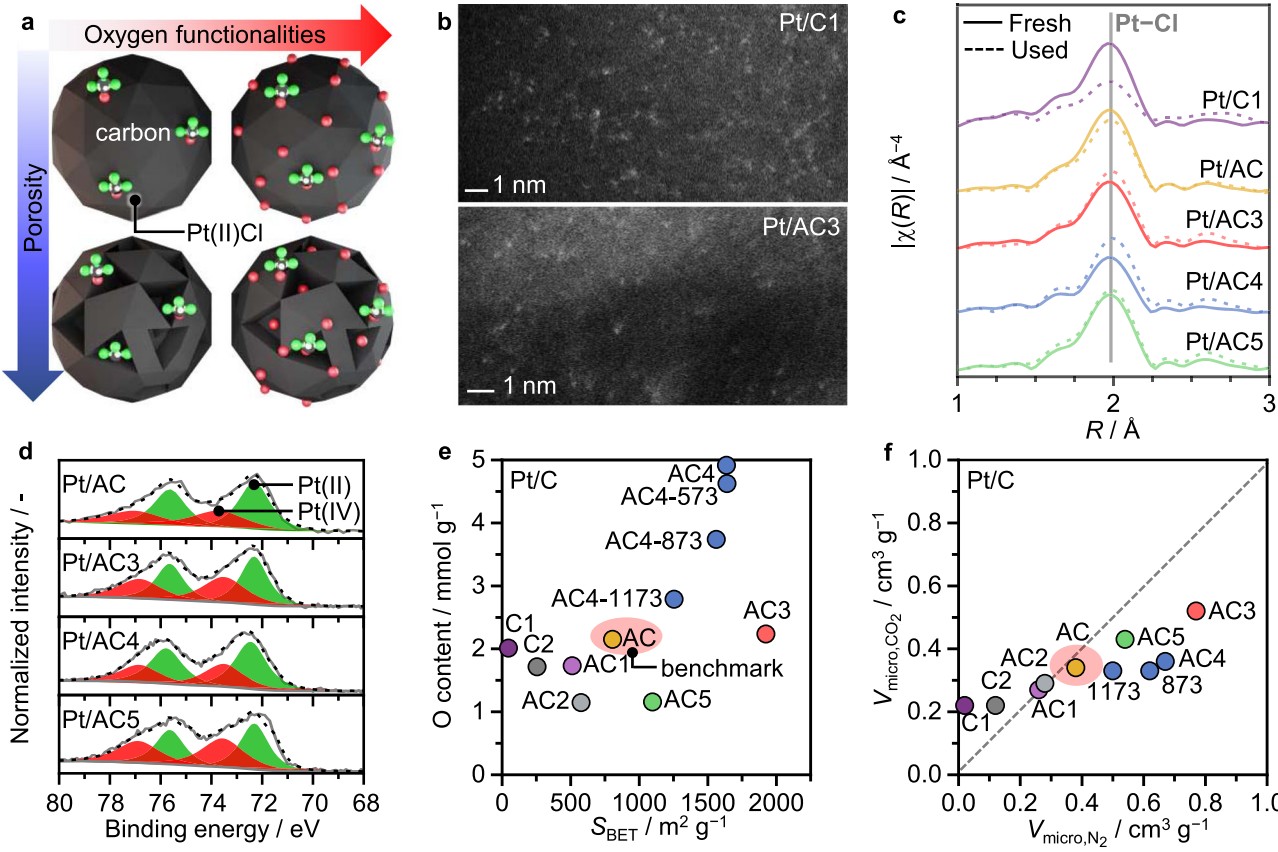

**Fig. 1 Preparation and characterization of the Pt/C catalysts. a** Synthetic strategy to control the porous properties and surface functionalization of the carbon support, while preserving the structure of the active platinum single-atom site in the catalytic material. **b** STEM, **c** Pt $L_3$ EXAFS, **d** normalized Pt $4f$ XPS, **e** overview of the total oxygen content, surface area, and **f** micropore volume of the Pt/C catalysts.

(Supplementary Table 4), while the ratio between C–O and C=O functionalities remained rather constant (~3:2 ratio of C–O to C=O groups) in all samples (Supplementary Table 9).

**The role of the support on the catalytic activity.** To evaluate the impact of porous properties and oxygen functionalities on the performance of Pt/C in acetylene hydrochlorination, all catalysts were studied under typical operating conditions (Fig. 2a, bed temperature, $T_{bed}$ = 473 K, HCl:$C_2H_2$ = 1.1:1, gas hourly space velocity based on acetylene GHSV($C_2H_2$) = 1500 h$^{-1}$). While vinyl chloride was the only product detected in all our catalytic tests (i.e., selectivity >95%), the activities varied significantly among the catalysts. Specifically, Pt/C1 and Pt/C2 showed similar acetylene conversion to the blank carbon supports (i.e., <3%, Supplementary Fig. 12), while their activated counterparts Pt/AC1 and Pt/AC2 ($T$ = 1123–1153 K, $CO_2$, Supplementary Table 1) reached comparable and even higher initial activities (expressed by the turnover frequency, TOF, after 15 min time-on-stream, TOS) with respect to Pt/AC. Since the oxygen contents, both total and associated with particular surface functionalities, remained in a comparable range after activation (or even decreased slightly), this parameter disqualifies as dominating activity descriptor. Further considering the insignificant difference in initial activity between Pt/AC4 and Pt/AC5, exhibiting respectively the highest and lowest oxygen contents in the series, it appears that changes in porous properties, rather than oxygen functionalities, dominate the activity trend, possibly through altering the adsorption capacities of the reactants. To assess this hypothesis, $C_2H_2$, HCl, and VCM TPD-MS were performed over selected catalysts (Supplementary Table 10). Interactions with

VCM were similar for all samples and generally weaker compared to HCl and $C_2H_2$. Notably, also the inactive samples Pt/C1 and Pt/C2 exhibit moderate adsorption capacities for both reactants at 303 K. However, at reaction temperature (473 K), acetylene interaction, as quantified by volumetric chemisorption, reduced to comparably low magnitudes as observed for the blank carbon supports, while a substantial increment was found for active Pt/C catalysts (Supplementary Tables 11 and 12, Supplementary Fig. 13a). Possibly, this observation relates to a lower kinetic barrier for $C_2H_2$ chemisorption on the surface of Pt/C1 and Pt/C2, leading to an adverse effect on the adsorption capacity with increasing temperature. In line with this explanation, the two inactive samples exhibit the highest acetylene adsorption capacity at 303 K, at the lowest temperature of desorption, indicating that the strength of acetylene interaction is comparatively weaker compared to the active catalysts. Overall, these observations suggest a minimum acetylene adsorption capacity as key activity descriptor, yielding a step-function-type correlation between initial TOF and acetylene adsorption capacity at 473 K (Fig. 2b). Upon exceeding a region of optimal $C_2H_2$ interaction, the initial activity stagnates and even decreases. This behavior, in line with the Sabatier principle[41], has also been observed over metal-free N-doped carbon catalysts in acetylene hydrochlorination[30]. Accordingly, to optimize acetylene interaction, the porous properties and the total oxygen content can be adjusted. As shown in Fig. 2c and Supplementary Fig. 13b there is a direct correlation between the acetylene adsorption capacity, as determined by volumetric chemisorption, and the amount of accessible micropores (reflected in an increase in surface area for most samples). Taking equilibrium and kinetic factors into account, the adsorption potential is maximized if the mean micropore size is

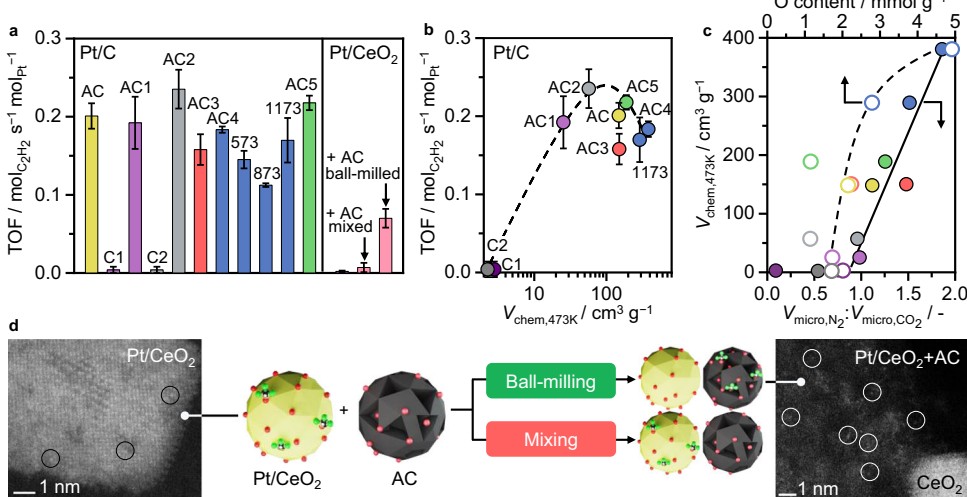

**Fig. 2 Activity descriptors for Pt/C catalysts in acetylene hydrochlorination. a** Initial activity, expressed as the turnover frequency, TOF. **b** Correlation between catalytic activity and acetylene adsorption capacity, as determined by volumetric chemisorption at reaction temperature, $V_{chem,473K}$. Error bars indicate the respectively lowest and highest activity obtained within three independent measurements. **c** Acetylene adsorption capacity as a function of the total oxygen content (open symbols) and the micropore accessibility, described as the ratio of the micropore volume, obtained from $N_2$ or $CO_2$ sorption (solid symbols). **d** Proximity effects between an active carbon support, AC, and the optimal metal nanostructure (Pt single atoms, circled) hosted on an inactive support (shown for $CeO_2$, but also applicable to the inactive carbon supports C1 or C2).

moderately larger than the probe molecule. Hence, acetylene, which exhibits comparable molecular dimensions to the selected probe molecules $N_2$ and $CO_2$ ($\sim 0.332 \times 0.334 \times 0.57$ nm) requires pore sizes of $\sim 0.7$ nm for optimal adsorption[42]. Secondary, thus less pronounced compared to the porous properties, also an increase in oxygen functionalities enhances the acetylene interaction (Fig. 2c).

To further substantiate the acetylene adsorption capacity as the central activity descriptor in acetylene hydrochlorination, a ceria-supported $PtCl_x$ SAC was prepared via incipient wetness impregnation[43]. Ceria was specifically chosen for its ability to effectively trap atomically dispersed Pt, thereby allowing to derive a metal site of comparable nature to those in the Pt/C series[7]. The presence of isolated Pt sites was confirmed by STEM and corroborated by XPS analysis (Supplementary Figs. 8 and 14). Similar to Pt/C1 and Pt/C2, also $Pt/CeO_2$ was inactive for acetylene hydrochlorination, in line with the relatively low quantity of acetylene chemisorbed at 473 K. More importantly, the acetylene adsorption capacity vanishes upon exposure to HCl, as a consequence of extensive bulk chlorination[44]. Remarkably, the acetylene interaction of carbon-based catalysts is virtually unaffected by HCl treatments, substantiating the unique suitability of carbon for this reaction (Supplementary Fig. 15) and revealing a key design criteria for the support for future directed studies.

To assess whether the $C_2H_2$ adsorption capacity of the inactive $Pt/CeO_2$ single-atom catalyst could be increased by physical mixing with AC, low-frequency ball-milling of a 1:1 mixture was performed (see Supplementary Methods for details). A slight increase in the initial activity upon mixing was found, which could be ascribed to the intrinsic activity of the carbon support. Remarkably, upon ball milling at higher frequency, a substantial increase in activity of the $Pt/CeO_2$-AC mixture was observed. This finding is well in line with a study of Ye et al., demonstrating the successful activation of inactive $Au/CeO_2$ for acetylene hydrochlorination upon ball milling with carbon, which has been ascribed to the improved acetylene adsorption capacity of the metal oxide/carbon-mixture[18]. However, based on the inactivity of the $Pt/CeO_2 + AC$ mixture, we can further conclude that proximity, achieved via mechanical mixing, is not sufficient to

obtain an active hydrochlorination catalyst. Instead, a high milling frequency is required to induce mechanical activation of the $Pt/CeO_2 + AC$ mixture, leading to a migration of Pt single atoms from $CeO_2$ to AC, as visualized by STEM (Fig. 2d). In the same manner, also Pt/C1 could be activated, while control experiments of ball-milling $Pt/CeO_2$ (or Pt/C1) in the absence of AC did not yield active catalysts. These results indicate that the Pt site must be directly bound to an "active" carbon support to effectively promote acetylene hydrochlorination.

**Catalyst deactivation and stability descriptor.** To assess the impact of the porous properties and oxygen functionalities of active carbon hosts on the stability of Pt/C catalysts in acetylene hydrochlorination, 12, 50, and 80 h tests were conducted under accelerated deactivation conditions (i.e., employing a high GHSV ($C_2H_2$) of 650–1500 h$^{-1}$, Fig. 3a, Supplementary Fig. 16). Evidently, the support strongly affects the stability of Pt/C in acetylene hydrochlorination, with Pt/AC3 and Pt/AC5 showing the best performance, even surpassing the benchmark Pt/AC. Particularly Pt/AC3 exhibits superior stability, maintaining a high productivity during a 50-h test (Fig. 3b, Supplementary Table 14).

By combining STEM, EXAFS, and XPS analyses, the preserved single-atom nature of Pt could be confirmed in the used catalysts, excluding deactivation due to metal nuclearity changes (Figs. 1c, 3c, d, Supplementary Figs. 1–5, 7, 8, Supplementary Tables 3–6). Further, all catalysts show a *ca.* 2–3 fold increase in the surface Cl content (Supplementary Table 4), which can be mainly ascribed to C–Cl contributions (Supplementary Table 6, Supplementary Fig. 9). On the contrary, the Pt–Cl coordination numbers as determined from EXAFS analysis remained similar, showing merely an increase in the second coordination sphere (i.e., Pt-Cl distance $\sim 2.9 \pm 0.03$ Å, Supplementary Table 3). Still, the progressive chlorination of the carbon support, particularly in proximity to the Pt atoms, leads to a reduced electron density at the metal sites, as concluded from a shift to higher binding energies in the Pt 4$f$ XPS spectra. A notable exception to this trend is found for Pt/C1, which shows a reduced Pt-Cl coordination number (i.e., from $3.1 \pm 0.3$ to $1.4 \pm 0.2$), possibly indicating limited HCl access as one reason for the inactivity of

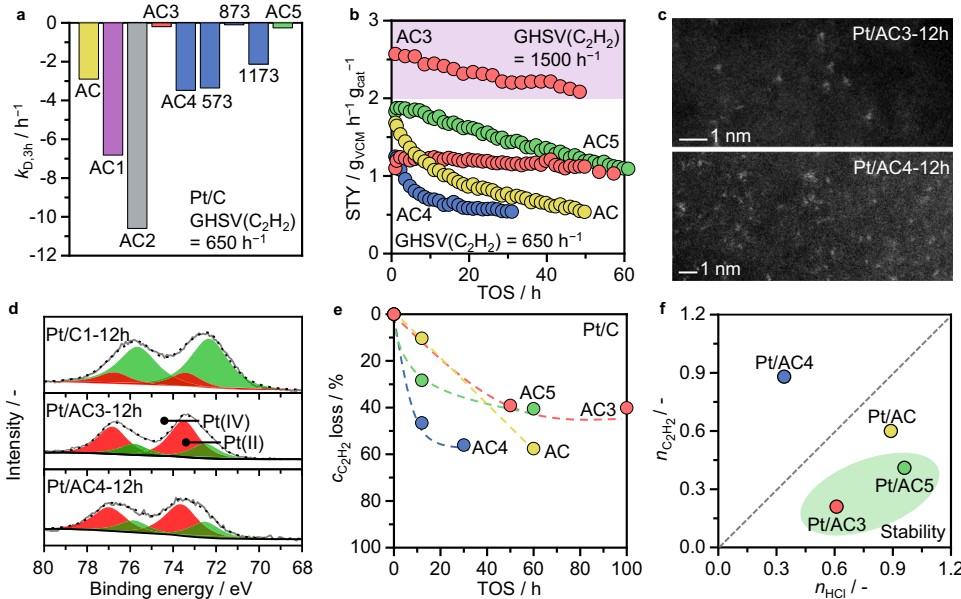

**Fig. 3 Stability and deactivation of Pt/C catalysts in acetylene hydrochlorination. a** Comparison of the deactivation constants, derived through linear regression of the data range within the first 3 h TOS (Supplementary Fig. 16). **b** Stability tests under accelerated deactivation conditions, using a GHSV ($C_2H_2$) of 650 $h^{-1}$ and 1500 $h^{-1}$ (colored area, AC3). **c** STEM and **d** normalized Pt 4$f$ XPS of used catalysts. **e** Decrease of the acetylene adsorption capacity, derived from TPD-MS, with TOS. **f** Partial reaction orders of HCl and $C_2H_2$.

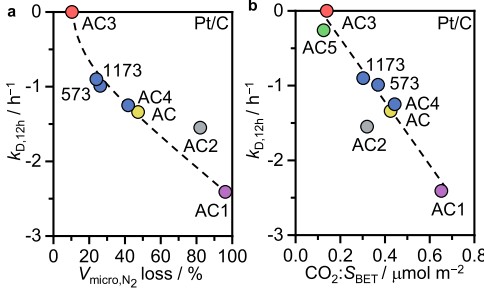

**Fig. 4 Stability descriptors for Pt/C catalysts in acetylene hydrochlorination.** Correlation between the deactivation constant and (**a**) the loss in total micropore volume, and (**b**) the density of acidic oxygen surface groups, estimated as the total content of oxygen functional groups evolving as $CO_2$ during TGA-MS, normalized by the surface area.

this sample. In this regard, the distinct deactivation rates of Pt/C can be directly correlated to the interactions of the support with the reactants and thus its intrinsic features, including changes in the porous properties and surface functionalities. With time-on-stream, the acetylene adsorption capacity gradually decreases for all catalysts, likely being the major cause for deactivation, with most severe effects on those catalysts with high partial reaction orders of acetylene (Fig. 3e, f, Supplementary Fig. 17). As the acetylene adsorption ability is a function of the accessible micropore volume (Fig. 2c), the relative loss of the latter through coke formation, especially beyond a certain extent, directly relates to the rate of catalyst deactivation (Fig. 4a, Supplementary Table 7). Comparative analysis of porous properties and oxygen surface functionalization of the fresh and used activated carbon supports and their corresponding Pt-containing catalysts was conducted to gain further insights into the factors governing coke formation (Supplementary Tables 7, 8). While in all cases, porosity and oxygen content decrease upon exposure to the reaction environment, the changes are more pronounced for the Pt/C catalysts compared to their respective supports. Still, the surface functionalization of the latter has a strong impact on the

overall coking activity of the Pt-based catalysts. Specifically, the rate of coking and consequently micropore blockage correlates well with the density of oxygen groups of acidic character (Fig. 4b). A possible explanation for the observed change from "inert" carbon to "active" carbon upon metal introduction is that carbon alone may effectively adsorb acetylene but cannot activate it further. Once the metal site activates acetylene, it is either transformed into vinyl chloride (over the metal site) or into coke precursors (over the acidic oxygen sites).

With coking identified as the main deactivation mechanism, TGA-MS of the fresh and used catalysts was performed to determine the quantity and type of the formed coke deposits. (Supplementary Fig. 18). Similarities between the two weight loss profiles of Pt/AC1 and Pt/AC3 suggest that a comparable amount of coke was generated in both samples (4–5 wt%), albeit leading to very different degree of deactivation. This result is in line with the similar quantities of acidic oxygen functionalities (Supplementary Table 8), but distinct porous properties, with Pt/AC3 having significantly enhanced micropore volume and surface area. (Supplementary Table 7). Overall, these results suggest the following activity and stability descriptors for the carbon support: high and accessible micropore volume (with pore sizes >0.7 nm) and a low content of acidic oxygen groups (Fig. 5a).

**Metal-specific interactions and descriptors.** Having identified the descriptors for the activity and stability of Pt/C catalysts in acetylene hydrochlorination, the general validity of these findings was assessed, by expanding the study to Au/C and Ru/C (see Supplementary Table 1 for catalyst notation and metal speciation). Following the previously described synthetic protocol, a series of Ru/C and Au/C catalysts was derived, employing water and acetone as impregnation solvents, respectively[20]. Combination of kinetic and volumetric chemisorption analyses revealed that alike Pt/C, also Ru and Au-based catalysts require a minimum of acetylene interaction at reaction temperature to yield active hydrochlorination catalysts (Supplementary Fig. 19, Supplementary Table 13). Accordingly, sufficient acetylene interaction could be verified as general activity descriptor for

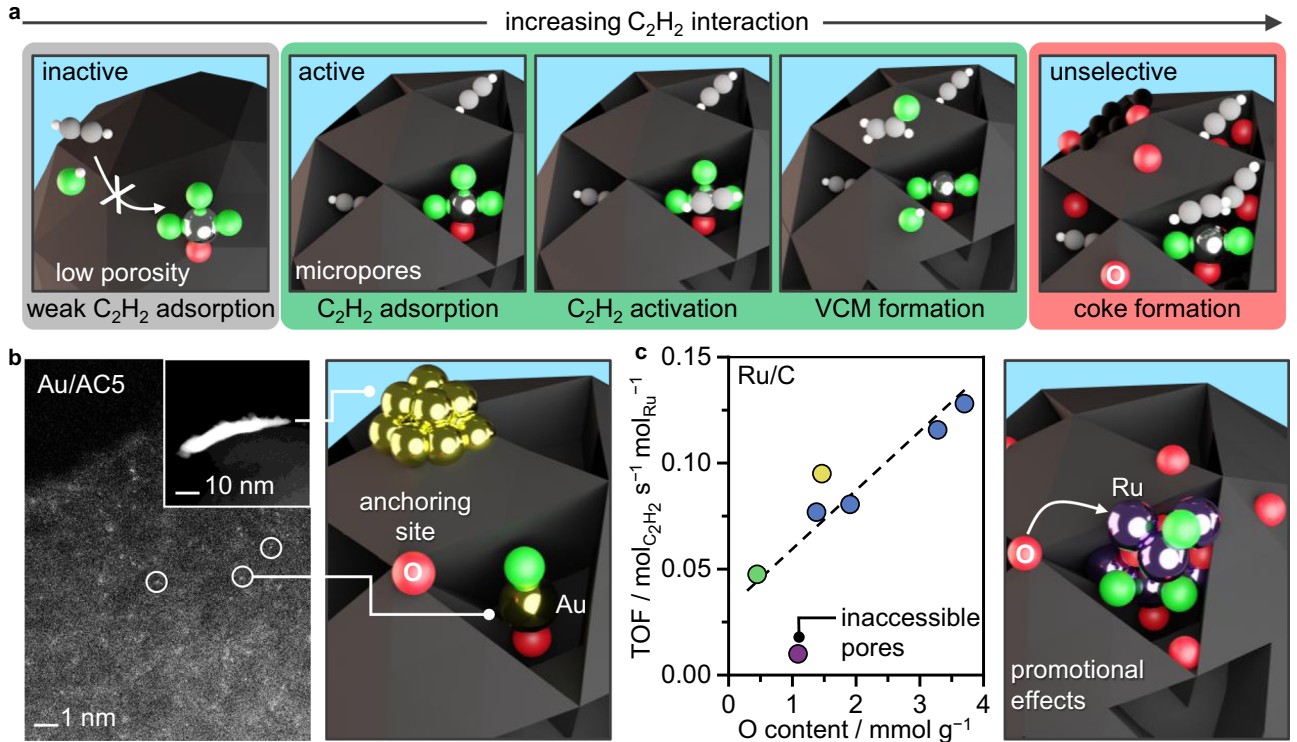

**Fig. 5 Metal-dependent role of the support. a** Mechanistic scheme of acetylene hydrochlorination exemplified over Pt/C catalysts as a function of increasing acetylene adsorption capacity, regulated by the porosity and oxygen content of the carbon support (gray: too weak, green: optimal, red: too strong). **b, c** Additional roles of the oxygen functionalities in determining the metal nanostructure of Au/C, exemplified by the STEM of fresh Au/AC5, and the initial activity of Ru/C catalysts.

metal-based hydrochlorination catalysts, explaining why supports like C1 or $CeO_2$ are impractical for this reaction. Notably, this correlation between activity and optimal acetylene interaction has also been identified for metal-free hydrochlorination catalysts, such as N-doped carbons (NC)[30]. To exclude the possibility that carbon, besides effectively adsorbing acetylene, may further serve as a carbon source in the catalytic cycle, [13]C labeled NC was synthesized from labeled aniline and tested under typical reaction conditions (see Supplementary Discussion for details)[29]. Using time-resolved MS analysis to track the formation of produced vinyl chloride ($C_2H_3Cl$) with $m:z$ 62, $m:z$ 63, and $m:z$ 64, carbon-atom exchange could be excluded (Supplementary Fig. 20), further substantiating that the key role of carbon in metal-based and metal-free systems is the promotion of acetylene adsorption in proximity to the active sites (i.e., metal nanostructure or N-functionalities).

Accordingly, we put forward that acetylene adsorption at the active metal sites is the central step that initiates the catalytic cycle. The latter requires direct proximity of Pt (or Au, Ru) to the carbon support. On the contrary, neither the carbon support itself, nor the metal site on a non-carbon support (or in a physical mixture with carbon) leads to sufficient acetylene supply (Supplementary Table 12). As Cl is already coordinated to the metal center, the reaction coordinate can directly advance as soon as acetylene is inserted in between the metal and Cl. This reduces the coordination and allows for the subsequent activation of HCl, followed by VCM formation through a H transfer. On the contrary, initial adsorption and activation of HCl is less favorable, as it leads to closing of the metal coordination sphere and hence compromises acetylene affinity (Fig. 5a).

Furthermore, for Au/C and Ru/C additional metal-sensitive factors come into play. STEM and XPS analyses revealed the selective formation of $RuO_xCl_y$ nanoparticles with an average

mean particle size of 1–3 nm over all carbons, while the Au nanostructure varied significantly (Supplementary Figs. 21, 22). Specifically, Au single atoms were predominantly present in Au/AC, while decreasing surface area and oxygen content favored the formation of larger gold nanoparticles (Fig. 5b). Accordingly, a key role of carbon is to provide sufficient surface area and suitable anchoring sites to ensure high dispersion and thus activity of Au. On the other hand, it has been shown that even those carbon supports which can promote atomic dispersion of Au (such as AC) fail to sufficiently stabilize Au under the harsh reaction conditions of acetylene hydrochlorination[14,23,35]. In this regard, further functionalization of carbon with additional O-anchoring sites (or other heteroatoms) merely leads to a shift in the predominating deactivation mode, from agglomeration to coking.

In the case of Ru/C catalysts, a direct correlation could be identified between the total oxygen content and the initial activity (Fig. 5c). This observation is well in line with previously described host effects of Ru catalysts, yielding twice the initial activity of Ru nanoparticles hosted on N-doped carbon (N-content *ca.* 10 wt%) compared to AC[24]. A similarly high initial activity could be obtained for Ru/AC4. However, alike Pt/C, also the deactivation rate of Ru/C catalysts evidences a direct correlation with increasing density of acidic oxygen groups (Supplementary Fig. 23). In this regard, there is a trade-off between the activity and stability of Ru/C catalysts in acetylene hydrochlorination.

## Discussion
Combining kinetic analysis and in-depth characterization over a comprehensive platform of carbons with varying porous properties and oxygen functionalities hosting Pt single atoms, Ru nanoparticles, and Au nanostructures, an optimum of acetylene interaction is identified as the general activity descriptor in

acetylene hydrochlorination, being finely tunable through adjusting the microporosity. In particular, the high chemical stability of microporous carbon and its affinity to organic compounds endows this support with a unique ability to effectively adsorb acetylene despite exposure to HCl atmosphere, in contrast to other types of host, as exemplified for ceria. Furthermore, the direct contact between the carbon support and the metal site is required, while proximity to an "acetylene reservoir" in a physical mixture will not result in catalytic activity. The direct participation of the carbon support in the reaction cycle as a source of carbon atoms could be excluded with $^{13}C$ labeling experiments. By taking advantage of the intrinsic stability of the active Pt single-atom sites on all carbon supports under reaction conditions, a direct correlation could be established between the rate of catalyst deactivation and the density of acidic groups, which likely promote undesired polymerization reactions, leading to accumulation of coke and consequently pore blockage, reducing the acetylene interaction. Overall, these results suggest the following stability descriptors for the carbon support: high and accessible micropore volume (with pore sizes >0.7 nm) and a low content of acidic oxygen groups, as further verified for Ru-based systems. In the case of the latter, a direct correlation between initial activity and total oxygen content could be identified, leading to a trade-off in the overall performance (Fig. 5c). Similarly, also for Au-based catalysts, a high density of oxygen groups is beneficial to promote the desirable dispersion of gold sites by providing anchoring sites. Considering these metal-sensitive activity and stability descriptors, Pt stands out as the most suitable candidate to further optimize towards high performance in acetylene hydrochlorination. In fact, the herein developed Pt/C catalyst with the lowest density of acidic oxygen groups in the carrier (Pt/AC3), exhibits unparalleled stability, surpassing the performance of their state-of-the-art systems under accelerated deactivation conditions and emphasizing the need for holistic design strategies, taking the metal site, its environment, and the support into account. In this regard, a promising future direction is the exploration of supported metal ensembles (e.g., alloy-based systems)[45,46] to further optimize acetylene interaction. Going beyond acetylene hydrochlorination, the herein derived strategy to disentangle metal-support interactions and gain insights into the specific roles of textural and chemical properties of carbon is generally applicable to any carbon-supported metal-catalyzed reaction and thus provides a valuable tool to derive descriptors for a wide range of catalytic applications.

## Methods

**Catalyst preparation**. All metal-based catalysts (nominal metal loading 1 wt%) were prepared via an incipient wetness impregnation method, employing the corresponding metal chlorides as precursors dissolved in deionized water (for Pt, Ru[14,24], or acetone (for Au)[20]. The obtained solutions were added dropwise to the different carbon supports and ceria and subsequently dried at 473 K (heating rate = 5 K min$^{-1}$ hold time 12 h, static air atmosphere) and 623 K (heating rate = 5 K min$^{-1}$ hold time 1 h, flowing air atmosphere), respectively. The obtained carbon series, termed Pt/C, Ru/C, and Au/C, exhibit distinct metal nanostructures, as summarized in Supplementary Table 1. Catalysts after use in acetylene hydrochlorination for 12-h time-on-stream (TOS) are labeled metal/C-12h. Further details on the catalyst synthesis and the preparation of the carbon supports are provided in the Supplementary Methods.

**Catalyst characterization**. Multiple characterization methods were employed to determine the composition, porosity, and structure of the fresh and used catalysts, as summarized in Supplementary Table 2. The carbon structure was analyzed by X-ray diffraction (XRD) and Raman spectroscopy. The surface oxygen content and speciation were determined by thermogravimetric analysis (TGA) in He coupled to mass spectrometry (MS). The acidity of the surface oxygen groups was probed via temperature programmed desorption (TPD) of NH$_3$. The porous properties of the samples were assessed by N$_2$ and CO$_2$ sorption at 77 K and 273 K, respectively. Catalyst surface composition, metal speciation, and dispersion were determined by X-ray absorption fine structure (XAFS) analysis, X-ray photoelectron spectroscopy (XPS), X-ray diffraction (XRD), and scanning transmission electron microscopy (STEM).

The interactions of the catalysts with the reactants and product were studied with C$_2$H$_2$-, HCl-, and VCM-TPD and static volumetric C$_2$H$_2$ chemisorption at the reaction temperature. All characterization techniques and procedures are detailed in the Supplementary Methods.

**Catalytic evaluation**. The hydrochlorination of acetylene was evaluated at atmospheric pressure in a continuous-flow fixed-bed reactor set-up, depicted in Supplementary Fig. 24 and further described in the Supplementary Methods. In a typical test, the catalyst (amount, $W_{cat}$ = 0.1 g for initial catalytic activity tests and kinetic tests, and 0.25 g for stability tests) was loaded in the quartz reactor and pretreated in He at 393 K for 30 min. Thereafter, a total gas flow, $F_T$ = 15 cm$^3$ min$^{-1}$, containing 40 vol% C$_2$H$_2$, 44 vol% HCl, and 16 vol% Ar, was fed into the reactor at bed temperatures, $T_{bed}$ = 453–483 K, employing a high gas hourly space velocity based on acetylene, GHSV(C$_2$H$_2$) = 650–1500 h$^{-1}$ to assess the catalysts under accelerated deactivation conditions. Quantification of reactants and products, determination of the yield of vinyl chloride, $Y$(VCM), reaction rate, $r$, turnover frequency, TOF, space-time-yield, STY, and deactivation constants, $k_D$, as well as the evaluation of carbon mass balances was conducted using the protocols detailed in the Supplementary Methods. All catalytic tests were performed in the absence of mass and heat transfer limitations, as detailed in Supplementary Discussion.

## Data availability

The authors declare that the data supporting the findings of this study are available within the article and its Supplementary Information file. Source data are provided with this paper.

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

## Acknowledgements

This work was supported by ETH research grant (ETH-40 17-1) and GV (PROMETEO/2018/076). The Scientific Centre for Optical and Electron Microscopy (ScopeM) at ETH Zurich is acknowledged for the use of their facilities.

## Author contributions

J.P.-R. conceived and coordinated all stages of this research. S.K.K. and I.S. synthesized the catalysts, contributed to their characterization, and conducted the catalytic tests. A.A.-P., M.C.R.-M., and M.A.L.-R. prepared the carbon supports and conducted porosity and surface chemistry analyses. F.K., A.H.C., and S.B. conducted electron microscopy, X-ray absorption spectroscopy, and X-ray photoelectron spectroscopy analyses, respectively. All authors contributed to the writing of the manuscript.

## Competing interests

The authors declare no competing interests.
