## [Peer Review File · Nature Communications]

Reviewer #1 (Remarks to the Author):

The manuscript presents a comprehensive study on catalysts for acetylene hydrochlorination with reactor testing and detailed catalyst characterization. The study compares Pt, Ru and Au catalysts supported on C and concludes that the Pt/C has performance advantages. The results are original and significant in the development of improved catalysts for acetylene hydrochlorination and similar chemistries as well as more broadly for metal catalysts supported on carbon. The manuscript can be further improved by addressing the following issues:

1. It would be helpful to clarify some contradictions in the Introduction. On the one hand, the text claims that C is inert. But on the other hand, it also claims that C has useful metal-support interactions and, perhaps, even its own catalytic activity. What do the authors want to claim: (1) C is inert and provides only anchoring sites for Pt and other metals, (2) C changes properties of the deposited metal or (3) C has its own active sites and, therefore, Pt/C is a bifunctional catalyst?
2. The Introduction claims that one of the major causes for catalyst deactivation is coking. It is then not clear why C is the selected support. Wouldn't it be better to use SiO₂ or a zeolite, which can be calcined to remove coke and regenerate the catalyst?
3. It would be useful to compare the concept of single-atom Pt catalysts in the manuscript to the concept of limiting the size of Pt ensembles (number of Pt atoms in an active site) with alloying. For example, it has been shown that addition of Sn changes acetylene adsorption on Pt when there are fewer than 3 adjacent Pt atoms. Please see: DOI: 10.1021/cs400198f and DOI: 10.1002/anie.201309043.
4. Could the authors clarify a possible reaction mechanism: does acetylene adsorb on C while HCl adsorbs on Pt? Or the other way around? Do the authors claim that all spectator species (mostly vinylidene and ethylidyne) from acetylene do not form on their Pt/C? For example, please see DOI: 10.1016/j.molcata.2004.04.015.
5. It would be useful to compare the developed Pt/C catalyst to catalysts without precious metals, for example La oxide for oxidative chlorination: DOI: 10.1021/ja066913w and DOI: 10.1007/s11244-006-0085-7. Similar catalysts were patented by The Dow Chemical for hydrochlorination.
6. The manuscript claims that "the density of acidic surface groups, which likely promote undesired polymerization reactions, leading to accumulation of coke and consequently pore blockage". First, can there be any acid groups rather than those on the surface? If not, then "surface" is redundant. Second, this conclusion again begs the question why not avoid C altogether and use a support without any acid groups, such as SiO₂ or a zeolite.
7. It would be useful for consistency to use either element names or element symbols. The current text is mixing both throughout the paper. For example, in the Abstract, carbon, platinum, gold and ruthenium are spelled out but then Pt is used as a symbol.

Reviewer #2 (Remarks to the Author):

The paper deals with a successful attempt to shed light on the role of carbons as support of Pt single-atom catalyst stability and efficiency in acetylene hydrochlorination reaction.

It has been done varying porosity and chemical functionalizations of carbon support, with the intention to maintain anchored Pt SACs.

The work is well researched, I find also the originality and the interest to the broad readers of the journal. Thus, I suggest publication after revisions detailed hereunder:

1. Figure 3d and Supplementary Figure 1 reporting STEM analysis, hardly confirm the preservation single-atom nature of Pt used as catalyst. It is not enough to exclude random deactivation. I would lower the tones related to that in main text. Metal nuclearity can change very easily. For each Figure it could be the case to better underline with circles what is considered Pt SACs and/or even enlarge picture size.
2. I found the manuscript too long and redundant in some sections, it should be shortened. Also it is written in a specialistic manner, not perfectly in line with the journal guidelines.
3. Considering the wider expertise of readers of the journal, according to me, references section is poor. A lot of paper related to study and Pt SACs on diverse platforms such as MOFs are lacking. Corma pioneristic work on heterogenous catalysis is underestimated. See for example Chem Rev. 2018, 118(10), 4981–5079 and Angew. Chem. 2018, 57(52), 17094-17099

Reviewer #3 (Remarks to the Author):

The paper reports the synthesis of single atom Pt catalysts on a range of commercial carbon samples with the aim to correlate characteristics such as porosity and O content with activity and stability towards the hydrochlorination of acetylene. The authors correlate the porosity and following this the acetylene absorption with the activity of the catalyst while the O content is related to the stability and is directly related to coking of the catalyst pores. While the paper contains an enormous amount of materials characterisation the work is a further optimisation of the groups recent publication on Pt single site catalysts for this reactions (Nature Catalysis volume 3, pages376–385(2020) and takes a similar approach to the groups study on metal free reactions (ACS Catal. 2018, 8, 2, 1114–1121). In addition the correlations made such as the polymerisation of acetylene on strong acid sites have been proposed many times previously in Au systems prepared in strongly oxidising and acidic conditions (J. Am. Chem. Soc. 2015, 137, 46, 14548–14557). The same observations regarding porosity have been made previously by the authors “Micropores are easily blocked during acetylene hydrochlorination, but meso- and macropores are structurally stable” (<https://doi.org/10.1002/cctc.201902331>). Due to this I do not recommend publication in Nature Communication as the catalysts systems are already established and the conclusions have been previously drawn.

Comments and Suggestions

The authors focus strongly on the properties of the carbon due to the fact that they are able to make single site Pt catalysts on all materials claiming that “the nature of the Pt sites is comparable” if the precursors, impregnation method and drying temperature are maintained. However there are clear differences in the ratios of Pt (II) : Pt (IV) which evolves during the reaction. How is his corelated with activity? How does the carbon influence the stability of the Pt(II) oxidation state.

The authors cite previous EXAFS data related to one sample which adds little to the understanding beyond this one sample.

The authors report and show that on impregnation of Pt the amount of acidic groups increases moderately while the amount of neutral/basic groups increases – what are the surface chemical processes which lead to the incorporation of more O in to the carbon during a aqueous impregnation?

The descriptions of the C₂H₂ TPD experiments is hard to interpret – was C₂H₂ dosed at two different temperatures? It is hard to interpret the results in the table when it includes both a C₂H₂ desorption temperatures below 473K and an acetylene amount chemisorbed at 473K.

Furthermore the authors report that the adsorption of acetylene is related to the samples performance by comparison of orders of reaction. Does the carbon alone undergo coke formation? What is the selectivity for these reactions? Which fraction of the reaction order exponent results from the carbon surface and which from the Pt? Is Pt required for the coke formation?

In some cases the activity or stability is ascribed to a global characteristics such as C₂H₂ adsorption capacity but the underlying reason for this is not explored – e.g. is C₂H₂ adsorption capacity related to total surface area? Pore structure? Surface functionality?

The design criteria that a good catalyst needs “high and accessible micropore volume (with pore size >0.7 nm)” – is the 0.7 nm derived from any reason beyond the choice of N₂ nad CO₂ as adsorption probe molecules?

Figure 2 - Errors should be includes on the TOF calculations especially when claiming a volcano-Sabatier type plot using the log scale in figure 2b – for example AC and AC3 have the same absorption capacity and approx. 0.5 difference in TOF – the same order as claimed as difference between AC1 and 1173 (two “extremes” among the activated carbons) and the peak of the suggested volcano.

Manuscript NCOMMS-21-06080 - Response to Reviewers

Comments in *blue* - Replies in black - Actions in **bold**

Indicated page, line, or figure numbers refer to the revised manuscript and/or supplementary information with changes highlighted

Reviewer #1

The manuscript presents a comprehensive study on catalysts for acetylene hydrochlorination with reactor testing and detailed catalyst characterization. The study compares Pt, Ru and Au catalysts supported on C and concludes that the Pt/C has performance advantages. The results are original and significant in the development of improved catalysts for acetylene hydrochlorination and similar chemistries as well as more broadly for metal catalysts supported on carbon. The manuscript can be further improved by addressing the following issues:

We warmly appreciate the positive feedback of Reviewer #1 and thank him/her for recognizing the originality and significance of our study in the field of acetylene hydrochlorination and beyond. The thoughtful comments prompted us to further clarify the aims and the scope of this work, further improving its impact. Each point is addressed below with a description of the actions taken upon revision.

1. It would be helpful to clarify some contradictions in the Introduction. On the one hand, the text claims that C is inert. But on the other hand, it also claims that C has useful metal-support interactions and, perhaps, even its own catalytic activity. What do the authors want to claim: (1) C is inert and provides only anchoring sites for Pt and other metals, (2) C changes properties of the deposited metal or (3) C has its own active sites and, therefore, Pt/C is a bifunctional catalyst?

Thank you for pointing out this ambiguity, which also relates to comment #4 by Reviewer #3. We have now **clarified in the Introduction (page 3, lines 43-46)** that the commonly used activated carbons are virtually inactive in acetylene hydrochlorination (vinyl chloride yield, $Y(\text{VCM}) < 2\%$) under operating conditions typical of metal-based catalysts ($GHSV(\text{C}_2\text{H}_2) = 650 \text{ h}^{-1}$ and $T_{\text{bed}} = 473 \text{ K}$). To substantiate this fact, we have **provided the initial activities of all activated carbons employed as supports in this study (page 10, lines 194-195, Supplementary Figure 12). Comparative analysis of porous properties and oxygen surface functionalization of the fresh and used activated carbon supports and their respective Pt-containing catalysts was conducted to gain further insights into the factors governing coke formation (Supplementary Tables 7,8).** While in all cases porous properties and oxygen contents decrease upon exposure to the reaction environment, the changes are more pronounced for the Pt/C catalysts compared to their respective supports. Hence, the bare supports can be described as comparably inactive in acetylene hydrochlorination. However, in combination with an active metal site, the carbon support becomes a key ingredient of the overall catalyst, as it *(i)* provides suitable metal anchoring sites (as determined by the respective metal-host interaction) and *(ii)* ensures acetylene supply to the active sites. The latter is critical for vinyl chloride formation, but also induces coke formation, particularly over acidic oxygen functionalities (**Figure 4b**). A possible explanation for the observed change from “inert” carbon to “active” carbon upon metal introduction is that carbon alone may effectively adsorb acetylene but cannot activate it further. Once the metal site activates acetylene, it may either be transformed into vinyl chloride (over the metal site) or into coke precursors (over the acidic oxygen sites). **We have now included this discussion in a new paragraph of the main manuscript (page 15, lines 298-309), alongside a mechanistic scheme (Figure 5).**

Besides metal introduction, the inert carbon matrix can also be activated by doping with heteroatoms such as N, P, and B, which results in substantial catalytic activity ($Y(\text{VCM}) = 10\text{-}15\%$ at $T_{\text{bed}} = 473 \text{ K}$ and $30\text{-}40\%$ at $T_{\text{bed}} = 573 \text{ K}$, see e.g., ACS Catal. 2018, 8, 1114). Especially N-doped carbons (NC) are commonly

employed metal-free hydrochlorination catalyst. However, the active nitrogen sites also promote extensive coking, thus leading to accelerated deactivation and consequently a trade-off between activity and stability, **as is mentioned in the revised Introduction (page 5, line 84).**

2. The Introduction claims that one of the major causes for catalyst deactivation is coking. It is then not clear why C is the selected support. Wouldn't it be better to use SiO₂ or a zeolite, which can be calcined to remove coke and regenerate the catalyst?

We recognize that this important aspect was not well highlighted in the original manuscript. In fact, carbon materials stand out as the only practically suitable supports for metal-based catalysts in acetylene hydrochlorination, **as is now mentioned in the revised Abstract and Introduction (page 2, lines 2-9, page 3, lines 35-38, page 4, lines 45-47).** Other carriers such as silica, alumina, zeolites *etc.* render much less active catalysts (see e.g., Ye *et al.*, Nat. Commun. 2019, 10, 914). As revealed in this contribution, the prime underlying reason for their unique suitability in this reaction originates from their high acetylene adsorption capacity in an HCl-rich environment. This is not the case for other typically employed supports, such as metal oxides, as exemplified by Pt/CeO₂ in **Supplementary Figure 15.**

3. It would be useful to compare the concept of single-atom Pt catalysts in the manuscript to the concept of limiting the size of Pt ensembles (number of Pt atoms in an active site) with alloying. For example, it has been shown that addition of Sn changes acetylene adsorption on Pt when there are fewer than 3 adjacent Pt atoms. Please see: DOI: 10.1021/cs400198f and DOI: 10.1002/anie.201309043.

Controlling the metal nanostructure is indeed a key aspect in the design of hydrochlorination catalysts. In our recent work (Nat. Catal. 2020, 3, 376-385), we assessed metal nuclearity and coordination effects of Pt on carbon and thereby identified the promising activity of Pt(II)-Cl_x single atoms in comparison to their nanoparticle-based analogs. Building on these insights, the focus of our present contribution is to precisely uncover the key role of the carbon support as a longstanding need in this field. Accordingly, our strategy is rooted on the preservation of the previously identified active metal site, while systematically altering the properties of carbon. **We have refined the Abstract and key parts of the manuscript to clarify the aims and scope of our contribution (page 2, lines 7-9, page 6, lines 98-103).** Further investigations on other ensembles, such as alloy-based systems, is beyond the scope of this work and might deserve a future dedicated study, **as is now referred to in the discussion (page 19, lines 397-400, refs. 54, 55).**

4. Could the authors clarify a possible reaction mechanism: does acetylene adsorb on C while HCl adsorbs on Pt? Or the other way around? Do the authors claim that all spectator species (mostly vinylidene and ethylidyne) from acetylene do not form on their Pt/C? For example, please see DOI: 10.1016/j.molcata.2004.04.015.

Thank you for this valuable suggestion. **We have now included a mechanistic scheme in Figure 5 of the main manuscript and complementary discussion (page 17, lines 340-349),** based on the new insights on the (metal-dependent) contributions of carbon obtained in this study. Accordingly, we put forward that acetylene adsorption at the active metal sites is the central step that initiates the catalytic cycle. The latter requires direct proximity of Pt (or Au, Ru) to the carbon support. On the contrary, neither the carbon support itself, nor the metal site on a non-carbon support (or in a physical mixture with carbon) leads to sufficient acetylene supply. **We have now included acetylene chemisorption data of the pure carbon supports in Supplementary Table 12 to substantiate this conclusion.** As Cl is already coordinated to the metal center, the reaction coordinate can directly advance as soon as acetylene is inserted in between Pt and Cl. This reduces the coordination of Pt and allows for the subsequent activation of HCl, followed by VCM formation through a H transfer. On the contrary, initial adsorption and activation of HCl is less favorable, as

it leads to closing of the Pt coordination sphere and hence compromising acetylene affinity. Following this reaction scheme, and considering the relatively high Cl:Pt ratio found in all catalysts and the absence of hydrogen in the reaction atmosphere, we do not expect significant formation of vinylidene and ethylidyne species, as described in the respective reference on the hydrogenation of acetylene. This conclusion is further substantiated by the fact that vinylchloride was the only product detected in all our tests by mass spectrometry analysis, **as is now stated in the revised manuscript (page 10, lines 191-193).**

5. It would be useful to compare the developed Pt/C catalyst to catalysts without precious metals, for example La oxide for oxidative chlorination: DOI: 10.1021/ja066913w and DOI: 10.1007/s11244-006-0085-7. Similar catalysts were patented by The Dow Chemical for hydrochlorination.

As mentioned in the response to comment #2, the acetylene hydrochlorination reaction relies on carbon-based supports. Nevertheless, we have assessed the performance of lanthanum-based catalysts (La_2O_3 , LaOCl , LaCl_3), typically employed in the oxychlorination of methane to methyl chloride elsewhere, and found that they are totally inactive in acetylene hydrochlorination.

6. The manuscript claims that “the density of acidic surface groups, which likely promote undesired polymerization reactions, leading to accumulation of coke and consequently pore blockage”. First, can there be any acid groups rather than those on the surface? If not, then “surface” is redundant. Second, this conclusion again begs the question why not avoid C altogether and use a support without any acid groups, such as SiO_2 or a zeolite.

Thank you, **we have now omitted the word “surface” for simplicity.** Further, we agree with the Reviewer’s point that the identification of a carbon-free support would be desirable in view of possible catalyst regeneration strategies. Following the design criteria for the support derived in this contribution, future efforts in this direction could target the identification of alternative materials which can maintain high acetylene capacity in an HCl-rich environment (*i.e.*, >40 vol.% HCl), **as is now mentioned in the manuscript (page 12, lines 243-244).** Alternatively, as showcased in our contribution, the structural and compositional diversity and tunability of carbon materials offers ample room to optimize the surface chemistry towards minimizing coking and thus enhancing the catalyst lifetime to an industrially acceptable level.

7. It would be useful for consistency to use either element names or element symbols. The current text is mixing both throughout the paper. For example, in the Abstract, carbon, platinum, gold and ruthenium are spelled out but then Pt is used as a symbol.

For clarity, we have unified the use of elemental symbols throughout the manuscript. Exceptionally, the full names of the elements were kept in the Abstract to improve the readability for the broader readership, as pointed out by Reviewer #2.

Reviewer #2

The paper deals with a successful attempt to shed light on the role of carbons as support of Pt single-atom catalyst stability and efficiency in acetylene hydrochlorination reaction. It has been done varying porosity and chemical functionalizations of carbon support, with the intention to maintain anchored Pt SACs. The work is well researched, I find also the originality and the interest to the broad readers of the journal. Thus, I suggest publication after revisions detailed hereunder:

We thank the Reviewer for the accurate assessment and for recognizing the significance and quality of our contribution. By addressing his/her constructive criticism, we were able to further strengthen the quality and impact of our study.

1. Figure 3d and Supplementary Figure 1 reporting STEM analysis, hardly confirm the preservation single-atom nature of Pt used as catalyst. It is not enough to exclude random deactivation. I would lower the tones related to that in main text. Metal nuclearity can change very easily. For each Figure it could be the case to better underline with circles what is considered Pt SACs and/or even enlarge picture size.

To further corroborate the conclusions drawn in the manuscript with respect to the preservation of the metal speciation in the catalysts, **we have now provided the EXAFS analysis of key samples (i.e., Pt/AC, Pt/C1, Pt/AC3, Pt/AC4, Pt/AC5, Figure 1c, Supplementary Figure 7, Supplementary Table 3, page 7, lines 127-131, Supplementary Information page 3, lines 66-85).** As further suggested by the Reviewer, **additional STEM images in enlarged form have been added to the Supplementary Information to improve the visualization of the metal sites (Supplementary Figures 1-5).**

2. I found the manuscript too long and redundant in some sections, it should be shortened. Also it is written in a specialistic manner, not perfectly in line with the journal guidelines.

To ensure the highest clarity to a broad readership, **we have shortened and/or rewritten the Introduction and Results sections, moving technical details to the Supplementary Information.**

3. Considering the wider expertise of readers of the journal, according to me, references section is poor. A lot of paper related to study and Pt SACs on diverse platforms such as MOFs are lacking. Corra pioneering work on heterogenous catalysis is underestimated. See for example Chem Rev. 2018, 118(10), 4981–5079 and Angew. Chem. 2018, 57(52), 17094-17099

Thank you for the constructive remark. **We have included the indicated studies in the Introduction as relevant pioneering work on Pt single-atom catalysts (page 6, lines 87-89, refs. 42, 43).**

Reviewer #3

The paper reports the synthesis of single atom Pt catalysts on a range of commercial carbon samples with the aim to correlate characteristics such as porosity and O content with activity and stability towards the hydrochlorination of acetylene. The authors correlate the porosity and following this the acetylene absorption with the activity of the catalyst while the O content is related to the stability and is directly related to coking of the catalyst pores. While the paper contains an enormous amount of materials characterisation the work is a further optimisation of the groups recent publication on Pt single site catalysts for this reactions (Nature Catalysis volume 3, pages376–385(2020) and takes a similar approach to the groups study on metal free reactions (ACS Catal. 2018, 8, 2, 1114–1121). In addition the correlations made such as the polymerisation of acetylene on strong acid sites have been proposed many times previously in Au systems prepared in strongly oxidising and acidic conditions (J. Am. Chem. Soc. 2015, 137, 46, 14548–14557) (how to react against this?). The same observations regarding porosity have been made previously by the authors “Micropores are easily blocked during acetylene hydrochlorination, but meso- and macropores are structurally stable” (<https://doi.org/10.1002/cctc.201902331>). Due to this I do not recommend publication in Nature Communication as the catalysts systems are already established and the conclusions have been previously drawn.

We thank the Reviewer for appreciating the breadth of our experimental efforts. Still, we are dismayed that he/she failed to acknowledge the novelty of our contribution, which does not lie in the mere optimization of our previously reported Pt-based catalyst, but rather the rationalization of descriptors for the design of the carbon support in metal-catalyzed acetylene hydrochlorination. We disagree that the conclusions have been already drawn since previous studies by us and many others have focused on a single carbon material as carrier, such as the Pt/C reference alluded by the Reviewer (Nat. Catal. 2020, 3, 376-385), precluding robust conclusions on the role of the support. As well recognized by Reviewers #1 and #2, we are considering here not only Pt but also Au and Ru and a wide variety of carbons of tailored porosity and surface O functionalities. Furthermore, the conclusions on porosity mentioned (ChemCatChem 2020, 12, 1922) were derived for metal-free N-doped carbons, comprising very different materials to those applied here. In fact, we advocate that (i) the lack of studies reporting an extensive set of materials and their corresponding characterization and (ii) the extrapolation of conclusions from different families of materials are major reasons why sustainable alternatives to mercuric chloride for acetylene hydrochlorination have not yet reached the right level of maturity for wide industrial application. Herein, we show that the dedicated design of the carbon support brings substantial benefits, enabling stable catalytic performance at a two-fold increased gas hourly space velocity compared to the state-of-the art Pt/C single-atom catalyst.

Contrary to previous studies in this direction, including the one indicated by the Reviewer (J. Am. Chem. Soc. 2015, 137, 14548) and references therein, we draw unequivocal conclusions on the role of the carbon support, as the metal site could be preserved and thus decoupled from structural and compositional changes in the carrier. In fact, previous strategies, building on variations in the impregnation solvent to alter the properties of carbon (water, nitric acid, aqua regia, etc.) primary resulted in metal nuclearity and/or coordination changes, which especially in the case of Au-based catalysts heavily affect the catalytic performance (see for example Malta *et al.* Science, 2017, 355, 1399), making it challenging to obtain reliable correlations. We believe this is indeed the main reason for the persistent controversies about the role of the carbon support in this reaction, which we could finally clarify thanks to the high stability of platinum single atoms on carbon. However, our conclusions apply to other active phases, as demonstrated for ruthenium and gold systems. **We have stressed these aspects in the revised Abstract (page 2, lines 5-10) and Introduction (page 5, lines 78-89 and page 6, lines 98-103).**

All other inquiries are addressed below point-by-point with the corresponding actions taken.

1. The authors focus strongly on the properties of the carbon due to the fact that they are able to make single site Pt catalysts on all materials claiming that “the nature of the Pt sites is comparable” if the precursors, impregnation method and drying temperature are maintained. However there are clear differences in the ratios of Pt (II) : Pt (IV) which evolves during the reaction. How is his correlated with activity? How does the carbon influence the stability of the Pt(II) oxidation state.

To further corroborate the conclusions drawn in the manuscript with respect to the preservation of the active sites, **we have now provided the EXAFS analysis of fresh and used Pt/C catalysts (i.e., Pt/AC, Pt/C1, Pt/AC3, Pt/AC4, Pt/AC5, Figure 1c, Supplementary Figure 7, Supplementary Table 3, page 7, lines 127-131, Supplementary Information page 3, lines 66-85).** In the fresh catalysts, Pt-Cl and Pt-O/C coordination numbers of $\sim 3 \pm 0.3$ and $\sim 1 \pm 0.3$, respectively were determined, indicating a comparable chemical nature and coordination environment of the active metal sites (**Figure 1c, Supplementary Figure 7, Supplementary Table 3**), which is in good agreement with the fairly comparable Pt(II):Pt(IV) ratios obtained by Pt 4f XPS analysis. Notably, small variations of the latter are within the expected error of this technique and/or the fitting procedure, considering that the varying properties of the carbon supports (i.e., conductivity) affect the analysis itself. Upon exposure to the reaction mixture, all carbon supports chlorinate, as evidenced by an increasing C-Cl contribution in the Cl 2p XPS spectra (**Supplementary Table 6, Supplementary Figure 9**). On the contrary, the Pt-Cl coordination numbers as determined from EXAFS analysis remained similar, showing merely an increase in the second coordination sphere (i.e., Pt-Cl distance $\sim 2.9 \pm 0.03 \text{ \AA}$, **Supplementary Table 3**). Still, the progressing chlorination of the carbon support, particularly in proximity to the Pt atoms, leads to a reduced electron density at the metal sites, as visible from a shift to higher binding energies in the Pt 4f XPS spectra. A notable exception to this trend is found for Pt/C1, which shows a reduced Pt-Cl coordination number (i.e., from 3.1 ± 0.3 to 1.4 ± 0.2), possibly indicating limited HCl access as one reason for the inactivity of this sample. **We have included this discussion in the amended manuscript (page 14, lines 275-289).**

2. The authors cite previous EXAFS data related to one sample which adds little to the understanding beyond this one sample. The authors report and show that on impregnation of Pt the amount of acidic groups increases moderately while the amount of neutral/basic groups increases – what are the surface chemical processes which lead to the incorporation of more O in to the carbon during a aqueous impregnation?

This is an important point. Accordingly, **we have performed additional EXAFS analysis, as now included in the revised manuscript (Figure 1c, Supplementary Figure 7, Supplementary Table 3, page 7, lines 127-131, Supplementary Information page 3, lines 66-85).** Regarding the changes in the surface functionalization of the carbon support upon metal impregnation, several points have to be considered. Firstly, the platinum precursor used in this work, H_2PtCl_6 , is a strong acid that undergoes fast hydrolysis in water, during which the chloride ions are exchanged by water molecules: $[\text{PtCl}_6]^{2-} + x\text{H}_2\text{O} \rightarrow [\text{PtCl}_{6-x}(\text{H}_2\text{O})_x]^{2+x} + x\text{Cl}^-$ (W.A. Spieker *et. al.* Appl. Catal. A 2002, 232, 219-235). During the impregnation of the carbon support, Pt(IV) is partially reduced to Pt(II) *via* surface redox processes, thereby altering the surface functionalization of carbon (e.g., Fraga *et.al.* J. Catal. 2002, 209, 355; Fierro *et. al.*, Langmuir 1994, 10, 750). During the subsequent drying step in air, the carbon surface likely undergoes additional oxidation reactions, leading to further oxygen incorporation. Consequently, all Pt/C samples show a characteristic increase in the total oxygen content upon impregnation (**Supplementary Table 7**), which is particularly enhanced for the initially most reduced carbons (lowest oxygen contents). **These additional explanations are now provided in the main manuscript (page 9, lines 166-173).**

3. The descriptions of the C₂H₂ TPD experiments is hard to interpret – was C₂H₂ dosed at two different temperatures? It is hard to interpret the results in the table when it includes both a C₂H₂ desorption temperatures below 473K and an acetylene amount chemisorbed at 473K.

Thank you for raising this point. To ensure the highest clarity, **we have revised the experimental descriptions of the acetylene TPD-MS experiments (Supplementary Information page 5, lines 113-116) and their representation in Supplementary Table 11.** Accordingly, the samples were saturated with acetylene at 303 K, purged with He (303 K), and subsequently heated up to 533 K to initiate the desorption. The C₂H₂ MS signal was then monitored in the temperature range between 303-533 K. To quantify the fraction of acetylene desorbing at temperatures >473 K (reaction temperature), we integrated (i) the whole data range between 303 K and 533 K (indicated as total acetylene adsorption capacity) and (ii) the fraction of the data range between 473 K and 533 K (referred to as acetylene adsorption capacity above 473 K).

4. Furthermore the authors report that the adsorption of acetylene is related to the samples performance by comparison of orders of reaction. Does the carbon alone undergo coke formation? What is the selectivity for these reactions? Which fraction of the reaction order exponent results from the carbon surface and which from the Pt? Is Pt required for the coke formation?

To answer these questions, we have **performed control experiments by exposing the blank carbon supports to the equivalent reaction conditions of their Pt-containing counterparts.** All carbon supports were virtually inactive (*i.e.*, vinyl chloride yield < 4%, **Supplementary Figure 12**). A detailed comparative analysis of porous properties and oxygen surface functionalization of the respective fresh and used catalysts (*i.e.*, C and Pt/C) indicates that the coking activity of the pure carbon supports is inferior in comparison to their Pt/C analogs. Taking further into account the stark differences between distinct Pt/C catalysts despite their common metal site, we conclude that the interplay between the surface functionalization of the carbon (*i.e.*, acidic oxygen functionalities) and the metal site is responsible for the overall coking activity, **as is now discussed in a new paragraph of the main manuscript (page 15, lines 298-309).** Besides coke formation, as indirectly assessed through nitrogen sorption and thermogravimetric analysis of the used catalysts, vinyl chloride was the only product detected in all our catalytic tests by mass spectrometry, **as is now stated in the main manuscript (page 10, lines 191-193).** As, the error in the carbon balance was <5% in all tests, the VCM selectivity was >95%.

5. In some cases the activity or stability is ascribed to a global characteristics such as C₂H₂ adsorption capacity but the underlying reason for this is not explored – e.g. is C₂H₂ adsorption capacity related to total surface area? Pore structure? Surface functionality?

Thank you for raising this relevant point. **We have now extended the discussion in the main manuscript (page 11, lines 224-232)** on the underlying structural properties determining the acetylene capacity of the carbon supports. As shown in **Figure 2c** and **Supplementary Figure 7b** there is a direct correlation between the acetylene capacity, as determined by volumetric chemisorption, and the amount of accessible micropores (reflected in an increase in surface area for most samples). Secondary, thus less pronounced compared to the porous properties, also an increase in oxygen functionalities enhances the acetylene interaction.

6. The design criteria that a good catalyst needs “high and accessible micropore volume (with pore size >0.7 nm)” – is the 0.7 nm derived from any reason beyond the choice of N₂ nad CO₂ as adsorption probe molecules?

As explained in the response to the previous comment, the pore size distribution is the key property that determines effective acetylene adsorption of the carbon support, as illustrated in **Figure 2c**. Taking

equilibrium and kinetic factors into account, the adsorption potential is maximized if the mean micropore size is moderately larger than the probe molecule. Hence, acetylene, which exhibits comparable molecular dimensions to our selected probe molecules N₂ and CO₂ (~0.332 x 0.334 x 0.57 nm, *i.e.*, Lin *et al.* J. Am. Chem. Soc. 2017, 139, 8022-8028) requires pore sizes of ~0.7 nm for optimal adsorption. **This important aspect is now highlighted with a new scheme (Figure 5) and accompanying discussion in the revised manuscript (page 12, lines 226-232, ref. 51).**

7. Figure 2 - Errors should be included on the TOF calculations especially when claiming a volcano-Sabatier type plot using the log scale in figure 2b – for example AC and AC3 have the same absorption capacity and approx. 0.5 difference in TOF – the same order as claimed as difference between AC1 and 1173 (two “extremes” among the activated carbons) and the peak of the suggested volcano.

Thank you for the comment. **We have now included the error bars, marking the respectively lowest and highest activity obtained in three independent measurements.**

Reviewer #1 (Remarks to the Author):

The manuscript has been significantly improved. The authors adequately addressed all the issues. The manuscript is now suitable for publication.